# Controlling the Transverse Magneto-Optical Kerr Effect in Cr/NiFe Bilayer Thin Films by Changing the Thicknesses of the Cr Layer

**DOI:** 10.3390/nano10020256

**Published:** 2020-02-01

**Authors:** Hisham Hashim, Mikhail Kozhaev, Pavel Kapralov, Larissa Panina, Vladimir Belotelov, Ivana Víšová, Dagmar Chvostová, Alexandr Dejneka, Ihor Shpetnyi, Vitalii Latyshev, Serhii Vorobiov, Vladimír Komanický

**Affiliations:** 1Department of Technology of Electronic Materials, National University of Science and Technology (MISIS), Moscow 119049, Russia; hh@science.tanta.edu.eg; 2Department of Physics, Faculty of Science, Tanta University, Tanta 31527, Egypt; 3Russian Quantum Center, Skolkovo, Moscow Region 143025, Russia; mikhailkozhaev@gmail.com (M.K.); p.kapralov@rqc.ru (P.K.); v.i.belotelov@ya.ru (V.B.); 4Prokhorov General Physics Institute of the Russian Academy of Sciences, Moscow 119991, Russia; 5Institute of Physics, Mathematics and IT, Immanuel Kant Baltic Federal University, Kaliningrad 236041, Russia; 6Institute of Physics, Czech Academy of Sciences, Prague 18221, Czech Republic; visova@fzu.cz (I.V.); chvostov@fzu.cz (D.C.); dejneka@fzu.cz (A.D.); 7Sumy State University, 2, Rimsky Korsakov Str., 40007 Sumy, Ukraine; i.shpetnyi@aph.sumdu.edu.ua (I.S.); serhii.vorobiov@gmail.com (S.V.); 8Institute of Physics, P.J. Šafárik University, 041 80 Košice, Slovakvladimir.komanicky@upjs.sk (V.K.)

**Keywords:** transverse magneto-optical Kerr effect (TMOKE), ferromagnetic coupling (FMC), suppression or enhancement of magneto-optical properties, magnetic nanostructure multilayers, reflectivity

## Abstract

Here, we demonstrate the impact of ferromagnetic layer coating on controlling the magneto-optical response. We found that the transverse magneto-optical Kerr effect (TMOKE) signal and TMOKE hysteresis loops of Ni_80_Fe_20_ thin layers coated with a Cr layer show a strong dependence on the thickness of the Cr layer and the incidence angle of the light. The transmission and reflection spectra were measured over a range of incidence angles and with different wavelengths so as to determine the layers’ optical parameters and to explain the TMOKE behavior. The generalized magneto-optical and ellipsometry (GMOE) model based on modified Abeles characteristic matrices was used to examine the agreement between the experimental and theoretical results. A comprehensive theoretical and experimental analysis reveals the possibility to create a TMOKE suppression/enhancement coating at specific controllable incidence angles. This has potential for applications in optical microscopy and sensors.

## 1. Introduction

Magnetic multilayered structures are of prime interest, as their properties significantly differ from the corresponding bulk materials. Nanostructured thin films are also of great technological importance for applications in nanoelectronics and spintronics [1,2], data storage technologies [3,4], magnetic and biological sensors [5,6,7] and optical filtering [8,9]. Many practically important physical effects have been discovered in multilayered films, including exchange coupling between ferromagnetic films separated by a non-ferromagnetic layer. 

Giant magnetoresistance in Fe/Cr multilayers and antiferromagnetic coupling between ferromagnetic layers assisted by the Cr-spacing generated particular interest in ferromagnetic film systems containing Cr layers. Controlling the properties of the layers and interfaces makes it possible to develop new concepts that would potentially result in novel applications [10,11,12,13]. In this paper, we apply ellipsometry and the magneto-optical Kerr effect (MOKE) as effective methods to characterize the ultra-thin bilayer films of NiFe coated with different thicknesses of Cr. It is demonstrated that the functional Cr layer at an optimized thickness and angle of incidence behaves as an enhancement layer. 

In general, MOKE is of interest for light modulation, with a magnetic field that typically requires MOKE enhancement by interferometric [14] or resonance [2,15] methods. On the other hand, magneto-optical effects provide a powerful nondestructive technique to investigate magnetization behavior in magnetic nanostructures, such as measuring magnetic hysteresis loops and imaging the magnetic domains [16,17]. The transverse magneto-optical Kerr effect (TMOKE) is a simple, economical and sensitive method to measure thin magnetic films in comparison with superconducting quantum interference device (SQUID) and Vibrating Sample Magnetometer (VSM) [18]. In TMOKE, one examines the changes in the intensity of the reflected light from (or transmitted light through) a magnetized material [19]. This effect arises because the complex refractive indices of the magnetized material are different for left- and right-circularly polarized light. 

Spectroscopic ellipsometry methods are well developed to measure the optical parameters and thicknesses of multilayers, with low losses. It is more challenging to characterize absorbing multilayers, as there is a correlation between the complex refractive index and the layer thickness [20]. For magnetic multilayers, it is useful to combine ellipsometry and MOKE methods [21]. In particular, a simple analytical model can be proposed for a generalized magneto-optical and ellipsometry (GMOE) scheme in the transverse magneto-optical configuration. GMOE allows for fitting the unique wavelength-dependent diagonal permittivity of every layer in the multilayer films, and to identify the off-diagonal elements responsible for the magneto-optical signals [22].

In this work, we have investigated the effect of the interface between ferromagnetic and antiferromagnetic materials in bilayer ultra-thin films by employing elipsometry measurements along with TMOKE. We have compared the experimental results with a theory based on the GMOE model of a ferromagnetic film coated with a non-magnetic layer. It has been revealed that the angular response of TMOKE for a simple structure of NiFe coated with a Cr layer is sensitive to the thickness of the Cr layer, which creates suppression/enhancement points. Such behavior is caused mainly by the interferences between the layers, as the optical properties of thin Cr layers (below 20 nm) are found to be typical of dielectric materials. This could be of interest for the development of optical microscopy [23] and sensing [24,25].

### Generalised Magneto-Optical and Ellipsometry (GMOE) Formalism

In the case of linear, with respect to the layer magnetization, MOKE, the phenomenological description for the multi-layered system is based on the matrix-form permittivity for magnetic layers. Considering the ellipsometry and TMOKE responses, it is convenient to use an extension of the Abeles characteristic matrixes [26]. 

Assuming that the plane of incidence is the (y, z) plane and the magnetization is in the film plane (along the *x*-axis for a transverse Kerr effect), as shown in Figure 1, the permittivity of an isotropic magnetic layer has the following form:(1)$$\hat{\varepsilon }=\varepsilon_{m}\left(\begin{array}{ccc} 1 & 0 & 0\\ 0 & 1 & -iQ\gamma \\ 0 & iQ\gamma & 1 \end{array}\right)$$
here, $\varepsilon_{m}$ is the diagonal permittivity, Q is the magneto-optical constant and γ is the x-directional cosine of the magnetization. In this case, the p-polarized ($H_{x}$, $E_{y}$ and $E_{z}$) and s-polarized ($E_{x}$, $H_{y}$ and $H_{z}$) waves remain the eigenfunctions of the wave equations.

The amplitudes of the corresponding fields at the zero boundary $\left(U_{0},\;V_{0}\;\right)$ are related to those at the end medium (U, V) via a characteristic matrix $\hat{M}$, as follows:(2)$$\left(\begin{array}{c} U_{0}\\ V_{0} \end{array}\right)=\hat{M}\left(\begin{array}{c} U\\ V \end{array}\right)$$

The matrix $\hat{M}$ of the multi-layered film is composed of the characteristic matrices of the individual layers:(3)$$\hat{M}=\underset{j=1,2}{\prod }{\hat{M}}_{j}$$

The reflection (*r*) and transmission (*t*) coefficients are found from the following:(4)$$\left(\begin{array}{c} 1+r\\ \left(-1+r\right)\cos\theta_{0}/n_{0} \end{array}\right)=\hat{M}\left(\begin{array}{c} t\\ -t\cos\theta_{s}/n_{s} \end{array}\right)$$
here, $\theta_{0},\;\theta_{s}$ are the angles of incidence and refraction, respectively, and $n_{0}$, $n_{s}$ are the refractive indexes of the medium of incidence and the end medium (substrate), respectively. The form of the characteristic matrices depends on the polarization. 

This formalism can be extended to the case of linear magneto-optical (MO) effects. A simple analytical expression is obtained for the ${\hat{M}}_{j}$ of the magnetic layer and the TMOKE configuration when the permittivity tensor is given by Equation (1) [27]. In the present work, we demonstrate that the model can be applied to describe the variations in the reflected intensity caused by the layer re-magnetization. The presence of magnetization modifies only the characteristic matrix of the p-polarized waves. The form of the wave equations in the magnetic layer does not change, but there is a modification in the relation between the components of magnetic and electric fields, as follows:(5)$$\frac{dH_{x}}{dy}=ik_{0}\varepsilon_{m}\left(iQ\gamma E_{y}+E_{z}\right)$$
(6)$$\frac{dH_{x}}{dz}=-ik_{0}\varepsilon_{m}\left(E_{y}-iQ\gamma E_{z}\right)$$

In Equations (5) and (6), $k_{0}=2\pi /\lambda$, λ is the wavelength in a vacuum. The characteristic matrix of the magnetic layer is of the following form:(7)$$\hat{M}=\left(\begin{array}{cc} \cos\left(\beta h\right)-\varsigma \;\sin\left(\beta h\right) & i\sin\left(\beta h\right)/q\\ i\;q\;\sin\left(\beta h\right) & \cos\left(\beta h\right)+\varsigma \;\sin\left(\beta h\right) \end{array}\right)$$

The parameters entering Equation (7) are as follows:(8)$$\beta =k_{0}n_{m}\;\cos\theta_{m}\;,\;\varsigma =Q\gamma \;\tan\theta_{m\;},\;q=\frac{\cos\theta_{m}}{n_{m}}\;,\;n_{m}=\sqrt{\varepsilon_{m}}$$

In Equation (8), $\theta_{m}$ is the refraction angle in the magnetic layer and h is the thickness of the magnetic layer. In a linear approximation with respect to ς, the determinant of the characteristic matrix is equal to 1. 

In the case of an s-polarized wave, the characteristic matrix corresponds to that of a non-magnetic layer, that is, the diagonal components of $\hat{M}$ do not contain the term proportional to Q, and in the off-diagonal components, the parameter q is replaced by $\cos\theta_{m}n_{m}$. Therefore, the formalism of the characteristic matrices makes it possible to calculate both the ellipsometry and TMOKE responses. The ellipsometry parameter ρ is defined as the ratio of the reflection parameters $r_{p}$ and $r_{s}$ of the p- and s-polarized waves, as follows:(9)$$\rho =\tan\;\psi \exp\left(i\Delta \right)=\;\frac{r_{p}}{r_{s}}$$

The TMOKE parameter δ(γ) is defined as the relative change in the intensity of the p-polarized waves upon magnetization, as follows: (10)$$\delta \left(\gamma \right)=2\frac{{\left|r_{p}\left(\gamma \right)\right|}^{2}-{\left|r_{p}\left(\gamma =-1\right)\right|}^{2}}{{\left|r_{p}\left(\gamma =1\right)\right|}^{2}+{\left|r_{p}\left(\gamma =-1\right)\right|}^{2}}$$

## 2. Materials and Methods

In our experimental measurements, we investigated bilayer thin films of Cr/Ni_80_Fe_20_, which were prepared by magnetron sputtering (ATC Orion 8 Sputtering Systems, AJA International, North Scituate, MA, USA) on glass substrates. The thicknesses of the layers varied from 2 to 20 nm, and from 10 to 20 nm for the Cr and Ni_80_Fe_20_ layers, respectively. To obtain the permalloy layer, the Ni_80_Fe_20_ target (from AJA International, North Scituate, MA, USA) was used. The sputtering rates were 0.13 nm/s for Ni_80_Fe_20_ and 0.28 nm/s for Cr (99.99% purity; AJA International, North Scituate, MA, USA). The sputtering rates were optimized for the material used. The films were prepared in the condition of an ultrahigh vacuum and a high-purity argon atmosphere, therefore, a lower sputtering rate did not affect the layer quality. The thickness was controlled by the quartz resonator method. We also controlled the concentration of components in the NiFe layer after deposition by using energy dispersive X-ray spectroscopy (EDS) (Oxford Instruments, Oxford, UK) analysis. The results show that the deviation from the specified concentration did not exceed 0.5% for all of the samples, which was within the accuracy of the EDS detector used. 

The spectroscopic ellipsometry measures ψ and Δ in Equation (9) both represent the elliptical polarization output state after the reflection of linearly polarized light at an oblique incidence off the film sample. They depend on the refractive indexes and thicknesses of the individual layers in the film. Therefore, the method makes it possible to determine the parameters for which a model describing the reflectance spectra of the film system must be built in order to theoretically obtain the values of ψ and Δ. Then, the optical parameters of the individual layers are deduced from fitting the experimental and modelled data [28]. The spectra of ψ and Δ were measured using variable angle spectroscopic ellipsometry (VASE; J.A. Woollam and Co., Nebraska, USA) at two incidence angles of 65° and 70°, working in rotating analyzer mode with an incident wavelength range from 300 to 1050 nm. The WVASE32 software package, containing the refractive index database for a large number of materials, was used as a simulator of a sample to fit and analyze the measured spectra in order to define the optical constants of the individual layers of Cr and NiFe when combined in a bilayer thin film, and to examine if these values depend on a particular layer thickness. In addition, a bare NiFe film with a thickness of 20 nm was inspected to accurately determine the optical constants of the individual layers of the films. The measured parameters were used in the GMOE model to quantitatively describe the observed magneto-optical response. 

A schematic illustration of the TMOKE setup is shown in Figure 2. The experimental measurements were carried out at room temperature. A laser diode with a wavelength of 780 nm and a power of 5 mW was used as the light source. Subsequently, the polarizer light was focused onto the sample, placed in alternating magnetic field with a maximum intensity of 50 mT. The reflected light beam from the surface of the sample was collimated with a lens, and then divided by a Wollaston prism into two beams with vertical and horizontal polarizations. A balanced photodetector was used for the signal measurement. Simultaneously, the signals from the electromagnet and balanced photodetector were recorded and analyzed. 

We also measured the reflectance at a zero field and maximum TMOKE signal, which is proportional to the relative change in the reflected light intensity at two opposite directions of the external magnetic field (corresponds to δ(1) in Equation (10)) for a range of angles of incidence from 10 to 70°.

## 3. Results and Discussion 

Spectroscopic Ellipsometry Characterization

Figure 3 shows an example of the measured ellipsometry parameter spectra for the bilayer Cr/NiFe film, along with the modelled spectra. The calculations are based on a two-layer model. There is an obvious consistency between the measured and the calculated angles of *ψ* and ∆ within the fitting model. The difference is lower than 1%, which ensures the reliability of the data for the optical parameters of thin layers used. 

Figure 4 shows the complex-valued permittivity spectra $\varepsilon =\varepsilon^{\prime }+i\varepsilon^{\prime \prime }$ for the individual layers constituting the film system, which were deduced by fitting the experimental ellipsometry angles and theoretical angles calculated within the two-layer model. For comparison, the data for the bulk materials (in the form of thicker films) are given. The permittivity data deduced from fitting for different layer thicknesses are consistent. However, there might be a significant difference between the obtained permittivity values of the thin layers and those known for the bulk materials [29,30]. In the case of the Cr thin film, the permittivity shows a “less” metallic behavior. In particular, the real part of the permittivity for the Cr layer shows positive values. This is consistent with the increase in resistivity with decreasing the Cr layer thickness [10]. We will further demonstrate that the obtained permittivity spectra accurately describe the observed optical and magneto-optical behaviors.

The obtained optical parameters $\varepsilon^{\prime }$ and $\varepsilon^{\prime \prime }$ from the ellipsometry measurements of the individual Cr and NiFe thin layers will be used in the GMOE model to obtain the theoretical angular dependence of light reflectance vs. NiFe magnetization and the maximal TMOKE signal of $\delta_{max}$ within a range of incidence angles.

The angular dependence of reflectance for the p-polarized laser light at a range of incidence angles from 10 to 70° is presented in Figure 5. We simulated the reflectance by using the permittivity values obtained from the ellipsometry data. The experimental and calculated data agree well (with an accuracy of less than 10%), which demonstrates that the obtained permittivity spectra for thin layers quantitatively characterize the optical behavior of the multilayered thin films. The difference may be related to a weak dependence of the layer permittivity on the thickness, as seen in Figure 4. The reflectance shows a reverse dependence on the incidence angle, and it increases alongside the Cr layer thickness. 

Figure 6 shows the maximal TMOKE signals $\delta_{max}=\delta \left(1\right)$ measured and calculated for the Cr/NiFe thin films as a function of the incidence angle for different thicknesses of the Cr layer. For modelling, the magneto-optical constant for NiFe was taken as Q = 0.0177 − i 0.0063 [31], and the optical constants of the Cr and NiFe layers were taken from our ellipsometry measurements. When increasing the incidence angle $\theta_{0}$, the value of $\delta_{max}$ changed value, but the angle of the zero signal strongly depended on the thickness of the Cr layer, as follows:$\;\theta_{0}$ changed from 40 to 60° when the thickness increased from 5 to 20 nm. At a fixed $\theta_{0}$ within this range of angles, it was possible to observe a changing TMOKE response as the Cr layer thickness increases, as follows: firstly, $\delta_{max}$ decreased going through zero, and then increased manifold. In particular, a strong increase of four to five times was observed for $\theta_{0}=45-50^{{}^{\circ}}$. The mechanism of TMOKE is related to the dependence of the boundary conditions for fields ($H_{x}$, $E_{y}$) on the x-magnetization of the ferromagnetic layer. The boundary condition also includes the magnitudes of the reflected/transmitted waves. Then, the interference effect of the Cr layer, which is characterized by a relatively large and positive real part of the permittivity, considerably influences the TMOKE signal. The theoretical results describe well the experimental data within the all-optical model. Some discrepancies may be related to the definition of the magneto-optical constant Q and the possible impact of ferromagnetic/antiferromagnetic coupling causing spin diffusion. Figure 6 also reveals the possibility of analyzing an in-depth profile of the ultra-thin bilayer films by the TMOKE technique, as the signal angular spectra demonstrate different depth sensitivities. 

The investigated thin films were mounted in the TMOKE configuration, as shown in Figure 2. The TMOKE signal is represented by the relative change in intensity *δ*(*γ*) in Equation (10), when the directional cosine *γ* varies in the presence of the external field. This constitutes the transverse hysteresis loops, which were measured at two different incidence angles of 30 and 50° in the presence of an in-plane magnetic field and are given in Figure 7. 

The maximal TMOKE signal $\delta_{max}=\delta \left(1\right)$, which should correspond to the film magnetization $M_{s}$ aligned along the magnetic field, is strongly affected by increasing the thickness of the Cr layer and the angle of incidence. For $\theta_{0}=30^{{}^{\circ}}$, $\delta_{max}$ and the apparent $M_{s}$ gradually increase with increasing the Cr layer thickness. Increasing $\theta_{0}$ to 5$0^{{}^{\circ}}$ results in a transformation in the hysteresis loops; the loop shrinks in a vertical direction, increasing the Cr layer thickness as $M_{s}$ decreases. With a further increase in the Cr layer thickness, the loops reverse and the value of $\delta_{max}$ starts to increase. This behavior reflects the change in the TMOKE sign, as the signal increases six times when the Cr layer thickness increases from 5 to 20 nm. Such an unusual behavior could be quantitatively explained within the all-optical model, with the layer optical parameters found from the ellipsometry measurements, as demonstrated in Figure 6b. The coercivity does not change, with the Cr layer thickness confirming that the quality of the interface is not affected by the increase in Cr layer thickness.

## 4. Conclusions

In summary, this work examined the influence of a top layer of Cr on the optical and magneto-optical properties of the bilayer films based on NiFe, demonstrating that the use of a functional Cr layer may enhance the magneto-optical response. A generalized model for ellipsometry and transverse MOKE analysis based on the extension of the Abeles characteristic matrixes was demonstrated to quantitatively describe the magneto-optical response. To fit the experimental and theoretical data, it was important to determine the optical parameters of the thin layers of Cr and NiFe, which could differ substantially from those of the bulk counterparts. The optical parameters of the individual layers were extracted from the ellipsometry measurements. Some additional contributions to the enhancement of the transverse MOKE response could also be due to the coupling between NiFe and antiferromagnetic Cr.

## Figures and Tables

**Figure 1 nanomaterials-10-00256-f001:**
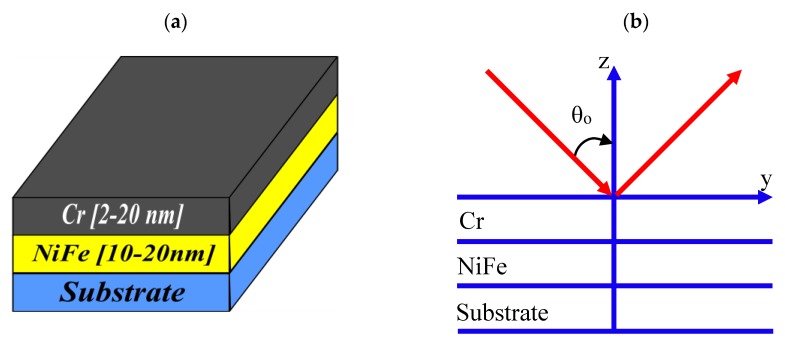
(**a**) Structure of the fabricated samples and (**b**) the optical scheme.

**Figure 2 nanomaterials-10-00256-f002:**
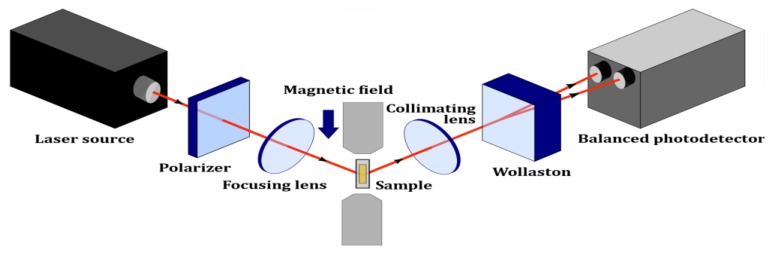
Schematic diagram of the transverse magneto-optical Kerr effect (TMOKE) setup.

**Figure 3 nanomaterials-10-00256-f003:**
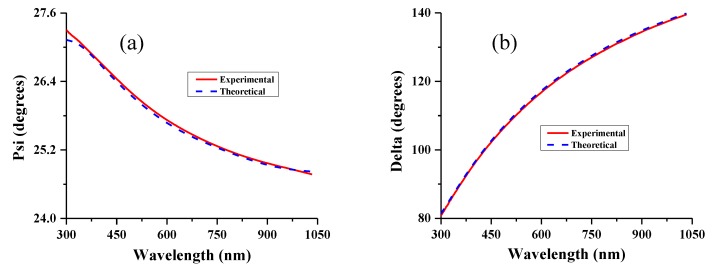
Experimental and calculated ellipsometry angles (**a**) for *ψ* and (**b**) for ∆; were obtained for Cr (2 nm)/NiFe (20 nm) films with a 70° incidence angle at room temperature. The measured and modelled values are represented with solid and dashed lines, respectively.

**Figure 4 nanomaterials-10-00256-f004:**
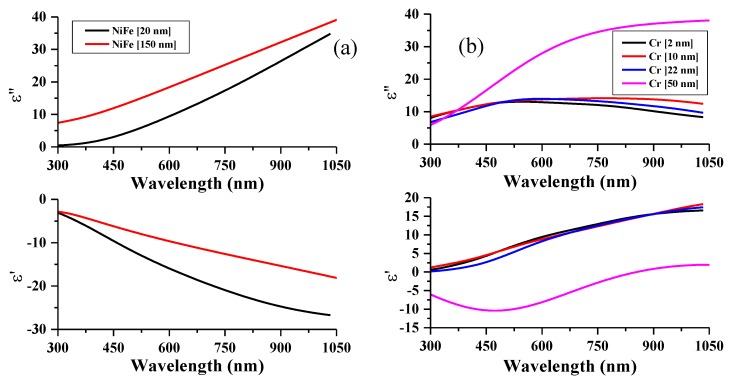
Real (*ε*′) and imaginary (*ε*″) parts of permittivity for (**a**) NiFe and (**b**) Cr layers obtained from fitting the ellipsometry measurement of a two-layer film system. For comparison, the data for bulk materials (in the form of thicker films) are given [29,30].

**Figure 5 nanomaterials-10-00256-f005:**
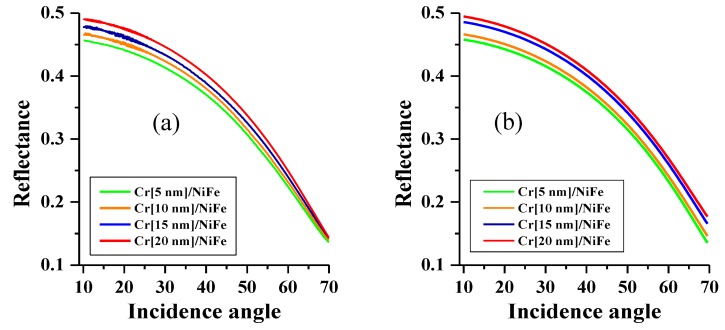
Reflectance parameters of p-polarized light vs. incidence angle $\theta_{0}$ for Cr/NiFe films with different thicknesses of the Cr layer and with the thickness of NiFe fixed to 10 nm: (**a**) experimental plots and (**b**) the theoretical plots calculated with the layer optical parameters deduced from the ellipsometry measurements.

**Figure 6 nanomaterials-10-00256-f006:**
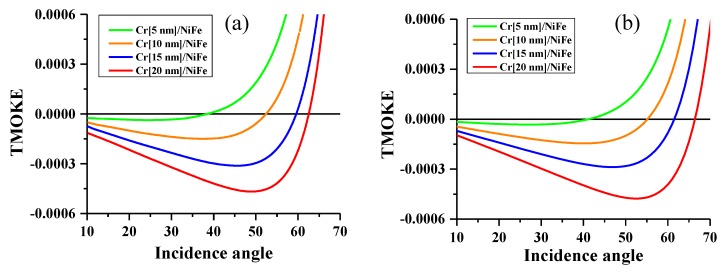
Maximal TMOKE signal $\delta_{max}$ vs. incidence angle $\theta_{0}$ for the Cr/NiFe films with different thicknesses of the Cr layer and with the thickness of the NiFe layer fixed to 10 nm: (**a**) experimental plots and (**b**) theoretical plots. The theoretical curves are obtained using the deduced results of optical parameters from the ellipsometry experiment.

**Figure 7 nanomaterials-10-00256-f007:**
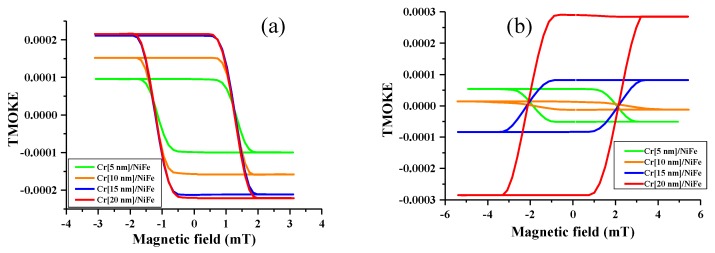
TMOKE hysteresis loops with an in-plane applied magnetic field at two different incidence angle Cr/NiFe films, with different thicknesses of the Cr film and the thickness of NiFe fixed to 10 nm: (**a**) at 30° and (**b**) at 50°.

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
