# Peer review of "Controlling the Transverse Magneto-Optical Kerr Effect in Cr/NiFe Bilayer Thin Films by Changing the Thicknesses of the Cr Layer"

_nanomaterials, 2020, doi:10.3390/nano10020256_

Round 1

Reviewer 1 Report

The paper concerns studies of Cr top layer influence on optical and magneto-optical properties of the Cr/NiFe bilayer thin films. The Authors investigated the transverse magneto-optical Kerr effect (MOKE). Hence, the topic is important because of the importance of magneto-optical materials, thin films for spintronics, data storage industry and even biological sensors. The object – thin films – also falls into the scope of Nanomaterials. The introduction is based on very recent references. It is only 21 papers but they seem well selected and cover the topics described in the manuscript. The experimental part presents preparation of Cr/NiFe/substrate with different thickness of Cr layers and subsequently, these films were characterized by spectroscopic ellipsometry, a method combining ellipsometry and MOKE. The Authors used a setup to perform TMOKE studies. The obtained results are interesting and might be considered for publishing in Nanomaterials. However, I have several questions and therefore I propose major revision.

The most important point: I would like to know what is the novelty and the biggest achievement of this work. Please, indicate clearly in your manuscript the novelty.

Page 2, line 47: “ellipsometric” please consider ”ellipsometry”

Page 2, line 67: “in ultra-bilayer thin films” please consider “in bilayer ultra-thin films”

Page 4, line 126-127: “measurements, we investigated bilayer thin films Cr (n)/Ni80Fe20 (10-20 nm) (n = 2-20 nm) were prepared”. Please rephrase because this sentence is not clear.

Page 4, line 129: You indicate different sputtering rates for Cr and NiFe layers. Do the differences in rates affect the quality of the layers?

Page 4, line 133: You indicated very 0.5% discrepancy between experimental and specified contents. It was established by EDS method which is excellent in qualitative but seems to be more semi-quantitative analysis (however for metals it gives relatively good results). Nevertheless, I would like to ask you how reliable are these results and if you tried also other methods.

Page 5, line 151: “After…” please consider “Subsequently…”

Page 6: On Figure 6 you show “relectance” or “reflectance”? The axis is described as angle. In caption you wrote about incidence angle. So, is it the incidence angle q0? Please, add values relating figures with mathematical equations. It would help follow your manuscript.

Page 7, 203: “from40°to 60°” should be “from 40°to 60°”

Author Response

Reviewer 1

Comments and Suggestions for Authors

The paper concerns studies of Cr top layer influence on optical and magneto-optical properties of the Cr/NiFe bilayer thin films. The Authors investigated the transverse magneto-optical Kerr effect (MOKE). Hence, the topic is important because of the importance of magneto-optical materials, thin films for spintronics, data storage industry and even biological sensors. The object – thin films – also falls into the scope of Nanomaterials. The introduction is based on very recent references. It is only 21 papers but they seem well selected and cover the topics described in the manuscript. The experimental part presents preparation of Cr/NiFe/substrate with different thickness of Cr layers and subsequently, these films were characterized by spectroscopic ellipsometry, a method combining ellipsometry and MOKE. The Authors used a setup to perform TMOKE studies. The obtained results are interesting and might be considered for publishing in Nanomaterials. However, I have several questions and therefore I propose major revision.

The most important point: I would like to know what is the novelty and the biggest achievement of this work. Please, indicate clearly in your manuscript the novelty.

Response on Novelty:

We made it clear the following novel features of the paper.

Combined elepsometry and MOKE investigations were performed for bi-layer film. This made it possible to model the MOKE response with actual layer optical parameters, which are quite different from those of bulk (or thick layers).

We have demonstrated that an angular dependence of the MOKE signal is affected by the thickness of the antiferromagnetic layer. Moreover, depending on the thickness, the signal changes sign and can be strongly enhanced.  This enhancement is similar to interferometry effect but obtained with conductive layers of Cr which could be also functional films for sensing applications.

Page 2, line 47: “ellipsometric” please consider ”ellipsometry”

We used “ellipsometry” instead of “ellipsometric” through out the text

Page 2, line 67: “in ultra-bilayer thin films” please consider “in bilayer ultra-thin films”

We used “in bilayer ultra-thin films” instead of “in ultra-bilayer thin films”

Page 4, line 126-127: “measurements, we investigated bilayer thin films Cr (n)/Ni80Fe20 (10-20 nm) (n = 2-20 nm) were prepared”. Please rephrase because this sentence is not clear.

The statement was changed. We simply wrote “The thicknesses of the layers varied from 2 to 20 nm, and from 10 to 20 nm for Cr and Ni80Fe20 layers, respectively”.

Page 4, line 129: You indicate different sputtering rates for Cr and NiFe layers. Do the differences in rates affect the quality of the layers?

The difference is sputtering rate might affect amount of impurities in the layer, lower deposition rate could result in higher amount of impurities deposited from the vacuum. In our case, the rates optimized for the materials were used. As the films were deposited at the condition of ultrahigh vacuum and high purity argon atmosphere, therefore, lower sputtering rates do not affect the layer quality.

We made this comment in the manuscript.

Page 4, line 133: You indicated very 0.5% discrepancy between experimental and specified contents. It was established by EDS method which is excellent in qualitative but seems to be more semi-quantitative analysis (however for metals it gives relatively good results). Nevertheless, I would like to ask you how reliable are these results and if you tried also other methods.

Approximate precision of EDS detector we used is in the order of 0.5%. So this discrepancy is within the experimental error which is stated in the manuscript. We have not tried other methods.

Page 5, line 151: “After…” please consider “Subsequently…”

We conceded “Subsequently” instead of “After”

Page 6: On Figure 6 you show “relectance” or “reflectance”? The axis is described as angle. In caption you wrote about incidence angle. So, is it the incidence angle q0? Please, add values relating figures with mathematical equations. It would help follow your manuscript.

Figure 5- the vertical axis represents the reflectance. We corrected this. We also added the used in equations symbol  for the angle of incidence for consistency. Similar changes were made in caption of Figure 6.

Page 7, 203: “from40°to 60°” should be “from 40°to 60°”

We corrected it.

Reviewer 2 Report

Bibliographic references should be supplemented especially for the less specialized readers of the journal.  The following references may be useful in this respect: https://doi.org/10.1016/j.sna.2019.04.003, https://doi.org/10.1016/j.cpc.2014.07.008, https://doi.org/10.1007/s10825-015-0698-9, http://dx.doi.org/10.1088/1361-6501/ab39b4, https://doi.org/10.1103/PhysRevB.100.214411

More explanations regarding the theoretical-experimental differences over ellipsometric angles are needed.

Some editing elements require revision. For example a section or chapter should not end with a figure or a table (must be surrounded by text), at least one explanatory text must follow them. Some figures should be restyled.

Author Response

Reviewer 2

Comments and Suggestions for Authors

Bibliographic references should be supplemented especially for the less specialized readers of the journal. 

Response:

We agree with the reviewer that additional references are needed for wider outlining the considered effects.  We included the proposed references with few more. 

More explanations regarding the theoretical-experimental differences over ellipsometric angles are needed.

In fact, the ellipsometry results are modelled with very high precision. The discrepancy is lower than 1%.  We made a comment on this.  Using the obtained values of layer permittivity, the reflectance angular spectra and MOKE signals were also modelled with high accuracy, although in this case a higher discrepancy is observed. This could be related with the dependence of the optical parameters on the layer thickness and the definition of the magneto optical constant.  We added this explanation.

Some editing elements require revision. For example a section or chapter should not end with a figure or a table (must be surrounded by text), at least one explanatory text must follow them. Some figures should be restyled.

We edited the text and the position of elements.

Reviewer 3 Report

The paper by Hashim et al. reports the influence of top layer of Cr on optical and magneto-optical properties of the bilayer films based on NiFe. The authors demonstrate that the use of Cr-layer may enhance the magneto-optical response. I have checked the manuscript and I would like to address a minor revision before it is accepted by Nanomaterials journal.

A comment on this are:

1) page 2 (Fig. 1(a)) and page 4 (lines 126 and 127)

I suggest omitting the values [5,10,15, 20 nm] and [10]. The significantly different values were collected in lines 126 and 127.

2) page 4 (lines 126 and 127)

Replace “Cr (n)/Ni80Fe20 (10-20 126 nm) (n = 2-20 nm)” by “Cr (2-20 nm)/Ni80Fe20 (10-20)”.

3) Page 5 and others, line 162 (please search the text)

I have some doubts about the following form “70á´¼ degrees”. I suggest” “70á´¼” or “70 degrees”.

4) Page 5, Fig. 3

I suggest to unify the colors of the lines, eg. experimental: red and theoretical: blue.

5) Page 6, Fig. 4

The x axis ranges are not equal. Should be the same, i.e. 1050.

6) Page 6, Fig. 4.

I do not quite understand for what systems the data are collected in this figure. Are they just layers only NiFe or Cr. I think this is poorly explained in Materials and methods. The different thickness values are given in this figure compared to what is in the text (lines 126-130).

7) Figs. 5-7

I don't understand why the authors change colors of lines for the respective system, for example Cr[5 nm]/NiFe: Fig. 5 – red; Figs. 6 and 7(a) – green; and Fig. 7(b) – black. This makes data analysis difficult.

8) Figs. 3 -7.

These figures are prepared carelessly:

the different axis thickness: between figures: Fig. 3 and others, and in the same figure: Fig. 5, the different size of fonts in legend – Fig. 7, the values at the ends and beginnings of the axis – Fig. 7, in Fig. 4 “(a)” and “(b)” are bold fonts, but in Fig. 5-7 “(a)” and “(b)” are normal ones, etc.

Author Response

Reviewer 3

Comments and Suggestions for Authors

The paper by Hashim et al. reports the influence of top layer of Cr on optical and magneto-optical properties of the bilayer films based on NiFe. The authors demonstrate that the use of Cr-layer may enhance the magneto-optical response. I have checked the manuscript and I would like to address a minor revision before it is accepted by Nanomaterials journal.

A comment on this are:

1) page 2 (Fig. 1(a)) and page 4 (lines 126 and 127)

I suggest omitting the values [5,10,15, 20 nm] and [10]. The significantly different values were collected in lines 126 and 127.

The relevant corrections were made.

 2) page 4 (lines 126 and 127)

Replace “Cr (n)/Ni80Fe20 (10-20 126 nm) (n = 2-20 nm)” by “Cr (2-20 nm)/Ni80Fe20 (10-20)”.

The statement was corrected.  We simply wrote, “The thicknesses of the layers varied from 2 to 20 nm, and from 10 to 20 nm for Cr and Ni80Fe20 layers, respectively”.

3) Page 5 and others, line 162 (please search the text)

I have some doubts about the following form “70á´¼ degrees”. I suggest” “70á´¼” or “70 degrees”.

We used 70 degrees.

 4) Page 5, Fig. 3

I suggest to unify the colors of the lines, eg. experimental: red and theoretical: blue.

We unified the colors of lines. Red for experimental and blue for theoretical.

 5) Page 6, Fig. 4

The x axis ranges are not equal. Should be the same, i.e. 1050.

We fixed the scale of X-axis.

 6) Page 6, Fig. 4.

I do not quite understand for what systems the data are collected in this figure. Are they just layers only NiFe or Cr. I think this is poorly explained in Materials and methods. The different thickness values are given in this figure compared to what is in the text (lines 126-130).

We made this explanation clearer. With the ellipsometry method, we determine the optical properties (complex refractive index or permittivity) of individual layers constituting the film system by fitting experimental and model results. Therefore, the optical parameters obtained by this way may differ for different film systems (with different layer thickness). We included an additional reference for the method [28] But our result show that they are quite consistent.  

7) Figs. 5-7

I don't understand why the authors change colors of lines for the respective system, for example Cr[5 nm]/NiFe: Fig. 5 – red; Figs. 6 and 7(a) – green; and Fig. 7(b) – black. This makes data analysis difficult.

We unified the color of these figures.

 8) Figs. 3 -7.

These figures are prepared carelessly:

the different axis thickness: between figures: Fig. 3 and others, and in the same figure: Fig. 5, the different size of fonts in legend – Fig. 7, the values at the ends and beginnings of the axis – Fig. 7, in Fig. 4 “(a)” and “(b)” are bold fonts, but in Fig. 5-7 “(a)” and “(b)” are normal ones, etc.

We fixed all the formats of figures.

Round 2

Reviewer 1 Report

It is a pleasure to see significantly improved manuscript. The results and experimental details were provided and the results got fairly better understandable. Therefore, I accept this paper in the current form. However, there are small mistakes in English. Therefore, I propose English revision if other faults exist in this manuscript.

page 4, line 136: "spattering" I suppose "sputtering"

page 7, line 207: "deference" should be "difference"